# Eco-Efficient Quantification of Glucosinolates in Camelina Seed, Oil, and Defatted Meal: Optimization, Development, and Validation of a UPLC-DAD Method

**DOI:** 10.3390/antiox11122441

**Published:** 2022-12-10

**Authors:** Salvador Meza, Yucheng Zhou, Jonathan Chastain, Yingying Yang, Hope Hua Cheng, Diliara Iassonova, Jason Rivest, Hong You

**Affiliations:** 1Eurofins Botanical Testing, US, Inc., 2951 Saturn St., Brea, CA 92821, USA; 2Cargill Global Edible Oil Solutions, 14305 21st Ave North, Minneapolis, MN 55447, USA; 3Eurofins US Food, 2200 Rittenhouse St, Des Moines, IA 50321, USA

**Keywords:** camelina, glucosinolates, extraction, liquid chromatography, internal standard

## Abstract

*Camelina sativa* (camelina) seed, oil, and defatted meal are widely used for food, animal feed, and other purposes. The accurate quantification of camelina glucosinolates is critical as their functionalities are highly dose-dependent. The classic quantification of glucosinolates in camelina products involves tedious desulfation steps, toxic reagents, and a lengthy instrument time because glucosinolates are easy to degrade and subject to interference in the liquid chromatography. Thus, we developed and validated an eco-efficient UPLC-DAD method for determining glucoarabin (GS9), glucocamelinin (GS10), and homoglucocamelinin (GS11) in camelina seed, oil, and defatted meal. Glucosinolates were extracted using 80% cold methanol to denature myrosinase, and were separated by an HSS T3 column without desulfation. Glucotropaeolin was used as an internal standard to track analyte degradation and loss during sample preparation. The method has shown high precision (relative standard deviations ranging from 4.12% to 6.54%) and accuracy (>94.4% spike recovery) for GS9-11, and all validation parameters passed the industry-consensus AOAC Appendix F criteria. To our best knowledge, this is the first eco-efficient and low-cost analytical method that is validated against strict AOAC criteria for the quantification of intact camelina glucosinolates. The method is suitable to be adopted as a new industrial testing standard to assist in the quality control of camelina products.

## 1. Introduction

*Camelina sativa* (camelina) is an oilseed crop in the *Brassicaceae* family [1]. There have been numerous reports on the emerging usage of camelina products as animal feed [2,3], functional food [4,5], biodiesel [6], and pharmaceutics [7]. The camelina seeds often undergo mechanical processing methods as camelina defatted meal is produced from the extraction of oils within. Camelina seed, oil, and defatted meal have been widely used for different applications and are, therefore, critical crop commodities for many industries [8].

Glucosinolates are secondary metabolites in *Brassicaceae* plants and share a general structure of glucose covalently bonded to a sulfur and a sulfated dioxamine with an R moiety of different linkages. The R groups are classified as aliphatic, aromatic, ω-methylthioalkyl, and heterocyclic (indole) [9]. Tissue compartmentalization has evolutionarily separated the chemically and thermally stable glucosinolates and their hydrolytic enzyme myrosinase, a beta thioglucosidase [10]. In the camelina plant, the glucosinolate profile has been examined, and the main glucosinolates are found to be the following aliphatic glucosinolates: 9-(Methylsulfinyl)nonyl glucosinolate (also known as glucoarabin or GS9), 10-(Methylsulfinyl)decyl glucosinolate (also known as glucocamelinin or GS10), and 11-(Methylsulfinyl)undecyl glucosinolate (also known as homoglucocamelinin or GS11) (Appendix A). Upon tissue disruption during mechanical processing, myrosinase can immediately react with glucosinolates. The hydrolytic degradants from glucosinolates are isothiocyanates (ITCs), which have the potential to induce anticarcinogenic effects by enhancing antioxidant potential and can ameliorate nonalcoholic fatty liver diseases by decreasing the level of lipid peroxidation in cells and tissues [11,12,13].

Although not all glucosinolates can be considered antioxidants [14], camelina glucosinolates GS9-11 have great potential to be further investigated for their antioxidative effects. This is because camelina seed and meal are believed to have strong antioxidative activity [15,16,17], and a positive correlation between the antioxidant activity and camelina glucosinolates was found in a recent study [18]. Moreover, the most predominant camelina glucosinolate GS10 is closely analogous to sulforaphane, a well-studied antioxidant isothiocyanate [19] for which the potency is possibly derived from the role of sulfenic acids and methylsulfinyl radicals as radical trapping agents [20]. Glucosinolates also indirectly possess antioxidant properties due to their metabolites formed from myrosinase activation, neutralizing free radicals and interfering with the Phase I and Phase II enzymes which trigger long-lasting antioxidant activity by decreasing the reactive oxygen species concentration and protecting the cell against DNA damage, respectively [21]. On the other hand, although glucosinolates are non-toxic when found in their intact forms, glucosinolate degradation leads to metabolites predominantly of the form thiocyanates, oxazolidinethiones, and nitriles that may cause liver and thyroid malfunctions [3,10]. Additionally, glucosinolates taste bitter and pungent to humans, chicken, fish, and pigs and can reduce feed intake and lead to growth retardation in finishing pigs when their content is higher than 15 µmol/g in canola meal [19]. However, based on our best knowledge, there is an overall lack of evidentiary support for the antioxidant, health, and toxicological properties of the specific GS9-11 found in camelina products. Therefore, an accurate but easy-to-use analytical method for camelina glucosinolates determination is critical to support academic research and product quality control for industry uses. This is because an underestimation of glucosinolate contents can put human and livestock health in danger, and an overestimation may lead to inaccurate interpretations of antioxidative research results for the potential health benefit exploration for GS9-11.

Accurate analysis of camelina glucosinolates is relatively difficult. Due to the complexity of secondary metabolites in camelina products, camelina glucosinolates has long been converted into a desulfated form with extensive enzyme treatment and clean-up steps for liquid chromatography analysis. For the analysis of rapeseed and canola glucosinolates, a classic glucosinolate analytical approach is recognized by both the International Organization for Standardization (ISO) [22] and American Oil Chemists’ Society (AOCS) [23], in which plant tissues are first freeze-dried to remove water from glucosinolate-containing tissues to prevent myrosinase-facilitated glucosinolate hydrolysis through thermal inhibition; then, the tissues are milled, immediately transferred to 70% methanol, and heated at 75 °C for 10 min to denature myrosinase; samples are subsequently desulfated into desulfoglucosinolates via a DEAE Sephadex column to remove impurities prior to the HPLC analysis. However, Oerlemans et al. [24] indicated that certain glucosinolates are heat sensitive and are significantly degraded at temperatures of ≥75 °C in less than 10 min. Doheny-Adams et al. [25] also carefully examined each of the sample preparation steps and found freeze-drying and heated methanol are not irreplaceable for the prevention of myrosinase-facilitated reactions; they found that these steps may actually lead to inaccurate glucosinolate results. The desulfation of glucosinolates requires a significant sulfatase reaction time, and the desulfation efficiency can be affected by feedback inhibition of the enzyme, which causes an incomplete desulfation of glucosinolates. In addition, the desulfation rate can vary within the column and yield results that will overestimate or underestimate the total glucosinolate concentration [25,26]. With the increased sensitivity and specificity that the modern liquid chromatography instrument and column can provide, direct measurement of intact glucosinolates may be feasible, and the desulfation steps can be omitted when the method is carefully designed [27,28].

For an intact camelina glucosinolate analysis for GS9-11, Berhow et al. developed an HPLC-UV method that utilized freeze-drying but successfully bypassed the desulfation procedures [27]. However, the instrument method requires as long as 40 min for each injection, making it inefficient for testing numerous samples on a routine basis. Yuan et al. [26] published a rapid method using a combined analysis with HPLC-ESI-MS/MS and proton NMR. It does not require corresponding GS9-11 reference standards, and the run time is only 16.5 min for the HPLC portion. While we were preparing for this manuscript, a simple and green ultrasound-assisted UPLC-HRMS/MS method was just published for GS9-11 determination in camelina meal [28], and the run time was only 11.5 min. The authors also used a response surface design to optimize the extraction conditions and utilized solid phase extraction to clean up the sample before injecting it into the instrument. However, tandem quadrupole MS, proton NMR, and high-resolution MS instruments are very expensive compared to liquid chromatography, making these methods difficult to adopt for routine product quality control in the food industry.

To avoid an underestimation of glucosinolate values because of incomplete desulfation and myrosinase-facilitated hydrolysis and to reduce unnecessary laboratory safety risks and labor costs, we developed an accurate and eco-efficient UPLC method with minimal preparation steps for extracting and quantifying GS9, GS10, and GS11 in camelina seed, oil, and defatted meal. Compared to other methods, we introduced and assessed an internal standard to ensure we can prohibit myrosinase-facilitated reactions and accurately track analyte degradation and loss during sample preparation. A validation study was then designed and conducted based on ICH guidelines [29]. The validation results were compared against industry-consensus AOAC Appendix F criteria [30]. To our knowledge, this is the first eco-efficient and low-cost analytical method that has been validated for the quantification of intact camelina glucosinolates.

## 2. Materials and Methods

### 2.1. Reagents, Chemicals, and Plant Materials

Glucoarabin potassium salt (GS9), glucocamelinin potassium salt (GS10), homoglucocamelinin potassium salt (GS11), and glucotropaeolin potassium salt reference materials were purchased from Extrasynthese (Genay Cedex, France). HPLC-grade methanol and acetonitrile were purchased from Fischer Scientific (Pittsburgh, PA, USA), and 85% phosphoric acid was purchased from Sigma Aldrich Inc. (St. Louis, MO, USA).

Study sample camelina seeds (growing conditions are currently undisclosable), oil, and defatted meal were provided by Cargill Global Edible Oil Solutions. Canola seeds (grown in a greenhouse in Saskatchewan, Canada in 2019), oil, and defatted meal were provided by Cargill Global Edible Oil Solutions and used as placebo samples for the validation study.

### 2.2. Preparation of Reference and Internal Standard Solutions

The reference standard solution was prepared by accurately weighing and transferring 5.0 ± 0.5 mg of each reference standard into a 25 mL volumetric flask. The flasks were then filled with approximately 15 mL of cold 80%/20% methanol/water and shaken with a wrist-action shaker for approximately 20 min until the standards were completely dissolved. The flasks were then filled to volume and inverted. A standard mixture working solution was created by transferring 1 mL of GS9, 7 mL of GS10, and 2 mL of GS11 into a 10 mL volumetric flask.

The internal standard stock solution was prepared by accurately weighing 10 ± 1.0 mg of glucotropaeolin potassium salt reference standard into a 25-mL volumetric flask and then filled with approximately 15 mL of 80%/20% methanol/water. The solution was then shaken for 10–15 min until completely dissolved, filled to volume, and inverted.

Calibration standard solutions were created to produce five-point calibration curves for each target analyte for GS9-11. Internal standard stock solutions have the same concentration across all levels in the final solutions. The process of making the final calibration standard solutions composed of standard mixture working solutions and internal standard stock solutions is presented in Table 1, but scaling up or down is acceptable. The five-point calibration curves were utilized for both quantification and method linearity determination.

### 2.3. Sample Preparation

Camelina seeds were grounded into powder with liquid nitrogen and stored in a −70 °C freezer. At extraction, they were further grounded to a finer powder using mortar and pestle and placed inside a box with dry ice to maintain low temperature and inhibit myrosinase activity. Approximately 0.5 ± 0.1 g of camelina seed fine powder was weighed and transferred to a 25 mL volumetric flask and immediately filled with 15 mL of cold 80%/20% methanol/water to denature myrosinase. Each sample solution was then spiked with 1 mL of internal standard stock solution.

Camelina oil and defatted meal samples were stored in a −20 °C freezer upon receipt. Approximately 0.5 ± 0.1 g of camelina oil or defatted meal was carefully transferred into a 25 mL volumetric flask and immediately filled with 15 mL of cold 80%/20% methanol/water. Each sample solution was then spiked with 1 mL of internal standard stock solution.

NOTE: When adjustment is needed, the concentration of the internal standard should be the same between the final sample solution and final calibration standard solutions.
(1)$$\mathrm{Amount}\mathrm{of}\mathrm{Internal}\mathrm{Standard}\mathrm{added}\mathrm{to}\mathrm{sample}\;(\mathrm{mL})=\frac{(C_{IFS}\times DF)\times Vsam}{C_{ISS}}$$
where:

*C_ISS_* = concentration of IS stock solution (mcg/mL);

*C_IFS_* = concentration of IS in final calibration standard solution (mcg/mL);

*V_sam_* = volume of total sample (mL);

*DF* = dilution factor of the sample.

The samples were then shaken for 1 h and filled to volume with 80%/20% methanol/water in the volumetric flask. Next, the solvents were transferred to centrifuge tubes and centrifuged for 5 min at 10,000 RPMs. The supernatant was then filtered through a 0.2-micron PTFE filter membrane and diluted 4× by transferring 0.25 mL of sample and adding 0.75 mL of extraction solvent in an LC vial. Samples were then analyzed using a UPLC-DAD.

When dealing with placebo samples for the accuracy study, the sample preparation procedures for canola seed, oil, and defatted meal were the same as for the corresponding camelina samples.

For any other similar type of crop sample, weighing amounts can be adjusted based on the sample’s expected level to target the mid-level calibration curve.

### 2.4. Column Selection and UPLC Analysis

Three UPLC columns (Acquity HSS C18 column 1.8 μm, 2.1 × 150 mm; Acquity HSS T3 column 1.8 μm, 2.1 × 150 mm; Acquity HSS PFP column 1.8 μm, 2.1 × 150 mm) with different column chemistries were selected for the column screen during the initial method development. The general chemical properties of these UPLC columns can be found on the Waters website https://www.waters.com/nextgen/us/en/products/columns.html (accessed on 16 November 2022).

The Acquity HSS T3 column, 1.8 μm, 2.1 × 150 mm (Waters, Framingham, MA, USA) was the most optimal column chosen for the final method and validation study. The instrument method information is listed in Table 2.

### 2.5. Single-Laboratory Validation Parameters

#### 2.5.1. Specificity

A method with adequate specificity can discriminate between compounds of interest and interferences. The method specificity test of each sample consisted of positive identification confirmation, negative result confirmation, and assay tests. Method specificity was demonstrated by running solvent blank, mid-level calibration standard mixture, camelina seed, oil, defatted meal, corresponding canola placebos (seed, oil, and defatted meal) that were expected to be free of target glucosinolates, and canola placebos that were spiked with reference standard mixture. The UV-Vis spectrum and retention time of each target analyte’s peak should be consistent between target peaks in the reference standard solution, camelina samples, and placebo samples spiked with calibration standards. No significant peaks should be presented in the chromatographic regions of interest in all three placebos and the blank solvent.

#### 2.5.2. System Suitability

The system suitability was demonstrated by injecting five replicates of mid-level calibration standards, which was performed for all GS9-11 and the internal standard. The % RSD (percent relative standard deviation) of peak area and retention time, tailing factor, and plate count were calculated for each glucosinolate. The % RSD passing criteria of retention time and peak area were 2.0% and 5.0%, respectively. The tailing factor should not be more than 2.0 for each analyte’s peak. The theoretical plate count should be no less than 10,000 for each analyte’s peak.

#### 2.5.3. Linearity

Five levels of calibration solution were injected at the beginning of each injection sequence. The calibration curves of the method covered the range approximately of 0.2–150 mcg/mL for GS9, GS10, and GS11. The approximate concentration range for internal standard glucotropaeolin was 1.0–4.0 mcg/mL. The correlation coefficient, calibration equation slope, and y-intercepts were automatically generated by the UPLC data processing software. Each calibration curve was made up of five data points, and the square of correlation of coefficient R^2^ for all curves must be ≥0.990.

#### 2.5.4. Limit of Detection (LOD) and Limit of Quantification (LOQ)

The LOD concentration was estimated by checking the detection limit of each analyte and its signal-to-noise ratio. The LOD signal-to-noise ratio should be >3:1. The LOQ concentration was estimated by checking the quantification limit of each analyte and the signal-to-noise ratio, with the lowest level calibration standard requiring a signal-to-noise ratio of >10:1.

#### 2.5.5. Precision

The method repeatability was determined by analyzing 6 separate weightings from a single composite of each matrix (camelina seeds, oil, and defatted meal) with the final method described above on Day #1 by Analyst #1.

For method reproducibility, intermediate precision tests were performed by running six samples for another two days. The full precision test was as follows: Run 1: Day#1, Analyst #1; Run 2: Day #2, Analyst #2; Run 3: Day #3, Analyst #1. A total of 18 tests were completed for each sample matrix.

#### 2.5.6. Accuracy

To demonstrate method accuracy, placebo samples of canola seed, oil, and defatted meal were spiked with two levels of GS9, GS10, and GS11 reference standards and the internal standard. The accuracy test was run in triplicate at each level using the final method described above. Recovery percentages were calculated by comparing the actual test results against the theoretical values (spike recovery testing).

#### 2.5.7. Robustness

The robustness test shows the reliability of an analysis with respect to deliberate variations in method parameters. Method robustness was determined by changing parameters (mobile phase composition, pH, flow rate, and column temperature) and comparing them to the original parameters according to the changed condition. Each original condition was changed one at a time at a difference of roughly ±5%, and each test was completed in 5 replicates. The column temperature was tested at 45 °C (original condition), 43 °C, and 48 °C. The flow rate was tested at 0.4 mL/min (original condition), 0.38 mL/min, and 0.42 mL/min. The pH and mobile phase composition were tested at 0.1% (original condition), 0.095%, and 0.105% phosphoric acid for Mobile Phase A. The percent result differences between each variation were reported.

### 2.6. Data Analysis

A linear curve was generated with a non-weighting linear regression curve using the ratio of internal standard area and component area, with × 100% as the ordinate (*y*-axis) and concentration of the internal standard, and final calibration standard 5-point curve solution as the abscissa (*x*-axis). The concentration of the sample was calculated from the regression analysis. The analyte concentration in the sample was calculated using Equation (2).
(2)$$Analyte\;in\;sample\;\left(\frac{mcg}{g}\right)=\frac{\left[sample,\frac{mcg}{ml}\right]\times Total\;Volume\;of\;sample\;\left(ml\right)}{Weight\;of\;Sample\;\left(g\right)}\;$$

These concentrations were calculated using linear regression and converted from their standard GS9 potassium Salt, GS10 Potassium salt, and GS11 potassium salts into their GS9, GS10, GS11 counterparts.

The UPLC chromatography software was Chromeleon C.D.S. (Thermo Fisher Scientific, Waltham, MA, USA). The mean, standard deviation, and relative standard deviation were calculated in Excel 2016 (Microsoft, Redmond, WA, USA). The comparison between internal standard and external standard calculated results was conducted using a paired Student t-test in Excel 2016.

## 3. Results and Discussion

### 3.1. Column Selection and Instrument Method Development

UPLC is an ideal instrument type for camelina glucosinolate analysis because it provides superior separation power, requires less organic solvent and run time, and is only 20–30% more expensive than HPLC. As the first step for developing a botanical analysis method on liquid chromatography, an instrument method with high specificity is usually established to provide accurate data to guide the optimization of sample preparation procedures [31,32]. Botanical analysis for industrial quality control applications requires the simultaneous separation of numerous phytochemicals within a short period of time [33,34,35]. For glucosinolate UPLC analysis in camelina products, the critical challenge is to separate an array of bioactive compounds such as glucosinolates, flavonols, lignans, phenolic acids, and nucleic acids, many of which have strong UV absorbance of around 225 nm (the maximum UV absorbance for GS9-11) and could potentially interfere with glucosinolates in the LC chromatogram (Figure 1).

In this study, three UPLC columns were selected to compare their separation performance (data not shown). All chromatograms were detected at 225 nm. The gradients were adjusted according to each column’s chemical properties. Waters Acquity HSS T3 column (1.8 µm, 2.1 × 150 mm) achieved the best separation performance, possibly due to its trifunctional bonding and end-capping process that allows for higher surface coverage to retain a wide range of polar and non-polar compounds. Subsequently, the gradient and other parameters were adjusted to provide baseline separations for all target camelina glucosinolates and the internal standard. The UV-VIS spectra of each target peak were checked to confirm there were no interferences. The final method provided adequate chromatographic separations (Figure 2).

Although we have validated this method using the Agilent brand UPLC instrument and Waters brand T3 column, we do not have concerns about the reproducibility of the method using apparatus from other brands (e.g., Sciex UPLC and PerkinElmer Neptune T3 column) due to our experience. A multi-lab validation study can be conducted in the future to further assess the reproducibility of the method in different laboratory conditions with different instruments.

### 3.2. Extraction of Camelina Glucosinolates and Internal Standard

When the plant or tissue is damaged, glucosinolates are degraded into their respective by-products in the presence of water and myrosinase [36]. Doheny-Adams et al. [25] highlighted that an 80% cold methanol extraction was sufficient to inactivate myrosinase and efficiently extracted glucosinolates without freeze-drying and heating methanol. Moreover, we found that it was impractical and unsafe to grind camelina seeds with liquid organic extraction solvent using a regular laboratory grinder or blender. To prevent myrosinase-facilitated reactions that occur instantly upon tissue breakdown, we maintained a low-temperature sample preparation environment before adding methanol to denature the myrosinase. For camelina and canola (placebo) seed analysis, liquid nitrogen was added during grinding, and samples were stored in a −70 °C freezer if not proceeded to the next steps immediately. If testing was then performed, the samples were placed into a container filled with dry ice to maintain the cold environment. The extraction solvent (80%/20% methanol/water) was always stored in a −20 °C freezer before use. The camelina oil, camelina defatted meal, canola oil, and canola defatted meal were kept in −20 °C freezers during storage.

When an analytical method appears to show forms of uncertainty in the sample preparation and instrument analysis, internal standard methods are useful to quantify analytes and account for those uncertainties. The greatest uncertainty within glucosinolate determination is their degradation rates influenced by the myrosinase enzyme, which hydrolyzes these phytochemicals into their respective by-products. When designed and executed appropriately, the amount of GS9-11 analytes lost in the camelina sample should be roughly proportional to the amount of internal standard loss because the internal standard will react with active myrosinase in a similar way. The proper choice of internal standard should satisfy requirements for a high resolution, stability, purity, structural similarity, and identical detector response with respect to the target analytes [37]. Therefore, a few glucosinolate candidates, such as sinigrin and glucotropaeolin, were spiked on camelina seed extracts to examine their elution (data not shown). Glucotropaeolin was finally chosen as the internal standard since it is commercially available, could be well separated from GS9, GS10, and GS11 by the final method, and did not co-elute with other compounds in camelina samples.

To further investigate the extraction efficiency of the final method and the validity of using glucotropaeolin as an internal standard, we conducted an assessment experiment by spiking reference standard solutions of GS9, GS10, GS11, and glucotropaeolin on the camelina seed sample before extraction occurred. The spiked and unspiked camelina seed samples were both tested in triplicate. The recovery rates of the targeted analytes were in the range of 75–125% (89.7% for GS9, 123.6% for GS10, 121.0% for GS11, and 84.7% for glucotropaeolin), showing a generally acceptable extraction efficiency and internal standard selection.

By analyzing the final validation study data, we compared the GS9-11 quantification results calculated according to the internal standard versus external standard using the paired Student t-test. The results showed that internal standard calculation provides significantly higher results for GS9, GS10, and GS11 in both camelina seed and defatted meal matrices (*p* < 0.01), highlighting the importance of including glucotropaeolin as an internal standard of the method.

### 3.3. Method Validation Results

#### 3.3.1. Specificity

The test results demonstrated adequate method specificity. The target peaks in the reference standard mixture, camelina samples, and spiked placebo samples had retention time differences of less than 5%, and their corresponding UV-Vis spectra were identical. Negative confirmation results confirmed no peaks in the retention time regions of interest in the solvent blank and canola placebos (Appendix A).

#### 3.3.2. System Suitability

The criteria (%RSD of retention time and peak area are no more than 2.0% and 5.0%; the tailing factor is more than 2.0; the theoretical plate count is no less than 10,000 for each analyte) for the system suitability parameters were all passed, with the raw data displayed in Appendix A.

#### 3.3.3. Linearity

The correlation coefficient R-squared value is greater than 0.990 for all analytes. The correlation coefficient, y-intercept, slope of the regression line, and residual sum of squares are provided in the Appendix A.

#### 3.3.4. LOD and LOQ

The method LODs were estimated as 0.090 mcg/mL, 0.056 mcg/mL, and 0.080 mcg/mL for GS9, GS10, and GS11, respectively. The LOQs were calculated to be 0.300 mcg/mL, 0.188 mcg/mL, and 0.266 mcg/mL with a signal-to-noise ratio of ≥10 for GS9, GS10, and GS11, respectively (Appendix A).

#### 3.3.5. Precision

In the precision test, the average concentration results, standard deviation (st. dev.), relative standard deviation (RSD), sample size, and confidence interval were reported. The repeatability %RSD_r_ and intermediate precision %RSD_R_ passing criteria were based on the AOAC Appendix F. GS9, GS10, and GS11 should have RSD_r_ percentages of no more than 5.3%, 3.7%, 3.7% and RSD_R_ no more than 8%, 6%, and 6%, respectively. The precision test results are shown in Table 3.

The RSD_R_ values of GS9, GS10, and GS11 in Camelina seed were 6.54%, 5.70%, and 4.15%, and the RSD_R_ values of GS9, GS10, and GS11 in Camelina defatted meal were 5.98%, 6.27% and 4.12%, respectively; thus, they were all within the acceptable range. GS9, GS10 and GS11 were not detectable in camelina oil with a chromatogram, as shown in Figure 2; therefore, method precision was not able to be determined for the camelina oil.

#### 3.3.6. Accuracy

The accuracy data are shown in Table 4. Per the AOAC Appendix F criteria, the expected %recovery depends on the analyte concentration. Although different levels of target analytes should have different passing criteria, a single 95–105% acceptable recovery range was selected as the pass criterion to make the assessment more straightforward.

Although glucosinolates in camelina oil were undetectable, an accuracy spike recovery test was performed to validate this matrix. The average recovery of GS9 is between 100.06 and 102.64% in camelina seed; between 99.31 and 101.11% in camelina oil; and between 100.00 and 103.00% in camelina defatted meal. The average recovery of GS10 is between 95.48 and 99.42% in camelina seed; between 100.90 and 104.49% in camelina oil; and between 97.43 and 104.07% in camelina defatted meal. The average recovery of GS11 is between 96.96 and 99.73% in camelina seed; between 96.16 and 99.75% in camelina oil; and between 96.66 and 103.02% in camelina defatted meal; these values were all within the acceptable range.

#### 3.3.7. Robustness

The data for robustness confirmation for all analytes are shown in Appendix A. The result %differences between the original and changed conditions were all <5% for mobile phase, column temperature, and flow rate.

To summarize, the developed method was validated using camelina seed, oil and defatted meal. Our method revealed high specificity and system suitability (<2% RSD on peak area; <0.5% RSD on retention time), great peak shape (tailing factor < 1.5; plate count >100,000) and linearity (R^2^ > 0.999), low detection and quantitation limits (LOD < 0.3 ppm; LOQ < 1 ppm), and high precision (RSDs ranging from 4.12% to 6.54%), accuracy (>94.4% spike recovery) and robustness (<1% difference between multiple testing conditions).

### 3.4. Result Comparison with Other Studies

In earlier studies, different quantification methods were established to determine the total concentration of glucosinolates in camelina seed and defatted meal. Although the glucosinolate concentration in camelina samples can vary due to genotype, geography, climate, storage, and processing conditions, accurate analytical methods should return results roughly of the same magnitude.

We determined the average total glucosinolate concentration to be 8.39 mg/g in camelina seeds (wet-basis results) and 15.95 mg/g in the defatted meal (Table 3). In a recent study, Pagliari et al. [28] reported 5.69 mg/g as the average total glucosinolate result in camelina-defatted meals using a rapid and green method, which is approximately 3-fold lower than our result. Matthäus and Zubr [38] reported the total content of glucosinolates to be in the range of 4.68–9.88 mg/g in camelina seeds and 7.54–12.17 mg/g in camelina defatted meal (oilseed cake). For camelina seeds, Berhow et al. [27] showed the total glucosinolate concentration to be 8–14 mg/g (dry-basis result), and Shuster et al. [39] obtained a result of 13.4 mg/g (dry-basis result). Yuan et al. [26] reported higher results (14.13 mg/g in camelina seeds and 31.81 mg/g in the defatted meal) using a ^1^H-NMR method for quantification.

Overall, our eco-efficient UPLC method could generate comparable results with other methods with much higher instrument costs or operation time, making it suitable to be adopted as a new industrial standard for glucosinolate determination in camelina products.

### 3.5. Pitfalls

One of the method’s drawbacks is the internal standard selection. The key differences of the chemical structure between the internal standard and the three target analytes are the benzene ring connected to the internal standard and the linear chain with the number of carbon atoms being greater than nine. The additional carbon atoms in a chain compared to the benzene can possibly cause variability in the enzyme pocket, making it more or less selective. The chain could increase the van der Waals interactions, or the benzene ring could increase pi-pi stacking in the binding region; therefore, although myrosinase is located inside camelina seeds, the myrosinase might not have a comparable effect on glucotropaeolin as it does for GS9, GS10, and GS11. In addition, the internal standard concentration is usually designed to be close to the target analytes in samples. However, the internal standard is generally on the lower end of the range in our method, mainly due to the expense of purchasing large amounts of the glucotropaeolin reference material. Although we conducted a spike recovery study (described in Section 3.2) to prove the validity of using glucotropaeolin as internal standard, alternative glucosinolates may be tested for future works. Moreover, a glucosinolate profile screening test could have been conducted using high-resolution mass spectrometry to identify whether there are any other major glucosinolates in camelina products, but we omitted this test because of the overwhelming amount of literature targeting only GS9-11 (accounting for over 95% of the total camelina glucosinolates).

## 4. Conclusions

To the best of the authors’ knowledge, we developed the first camelina glucosinolate quantification method that is both eco-efficient and able to pass the AOAC validation criteria of high specificity, linearity, precision, accuracy and robustness. Using this method, myrosinase hydrolysis was successfully inhibited by 80% cold methanol. Instead of being converted into desulfoglucosinolates via sulfatase solution and further cleanup procedures, intact glucosinolates in camelina products were separated from other interfering compounds using a T3 column. Moreover, glucotropaeolin was carefully selected as the internal standard to track and correct glucosinolate degradation during the whole process. Due to the performance and low-cost nature of running this method, we believe it is suitable to be adopted as a new industrial testing standard to assist in the quality control of camelina products.

## Figures and Tables

**Figure 1 antioxidants-11-02441-f001:**
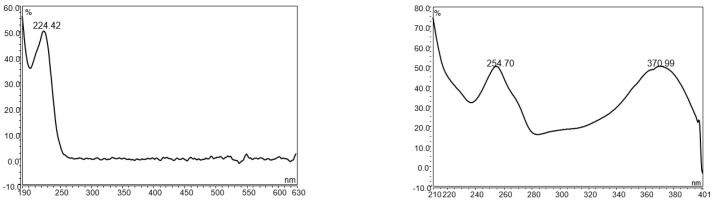
The UV spectrum of glucoarabin (**left**) and quercetin (**right**).

**Figure 2 antioxidants-11-02441-f002:**
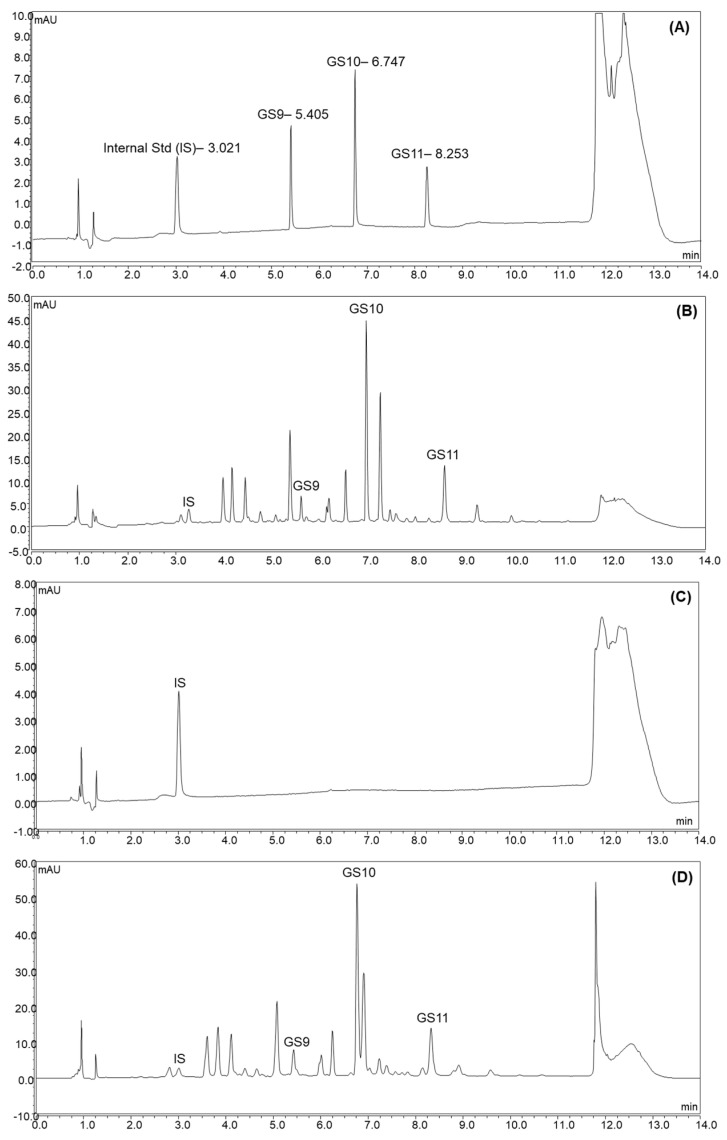
The chromatograms of reference standard mixture of GS9, GS10, GS11, and IS **(A**); extracts of camelina seed (**B**); camelina oil (**C**); camelina defatted meal (**D**) spiked with IS.

**Table 1 antioxidants-11-02441-t001:** An example of making final calibration standard solutions with GS9, GS10, GS11, and IS.

Calibration Std. Solution	Volume of Std. Mixture Working Solution (mL)	Volume of Internal Std. Stock Solution (mL)	Final Volume (mL) ^1^
Standard Level 1	0.02	0.01	1
Standard Level 2	0.04	0.01	1
Standard Level 3	0.1	0.01	1
Standard Level 4	0.2	0.01	1
Standard Level 5	0.9	0.01	1

^1^ 1-mL LC vials or volumetric flasks (if making more solutions) can be used as containers for the final solutions. Different amount of 80%/20% methanol/water was added to each container to fill to the final desired volume.

**Table 2 antioxidants-11-02441-t002:** The final instrument method.

UPLC System	Agilent 1290 Infinity II LC System		
Column	Waters Acquity HSS T3 column, 2.1 × 150 mm, 1.8 µm		
Column Temperature	45 °C		
Detection (Wavelength)	Diode Array Detector at 225 nm		
Injection Volume	2 µL		
Flow Rate	0.4 mL/min		
Run Time	14 min		
Gradient Program	Time (minutes)	% Mobile Phase A (0.1% phosphoric acid in water)	% Mobile Phase B (100% acetonitrile)
	Initial	92.0	8.0
	5.0	78.5	21.5
	7.0	77.0	23.0
	10.0	72.0	28.0
	10.2	20.0	80.0
	10.8	20.0	80.0
	11.0	92.0	8.0
	14.0	92.0	8.0

**Table 3 antioxidants-11-02441-t003:** Precision data of Camelina Seed and Camelina Defatted Meal. Camelina oil data are not shown because their GS9-11 levels were non-detectable.

	Result (mg/g)							
	**Camelina Seed**				**Camelina Defatted Meal**			
Compound	GS9	GS10	GS11	Total	GS9	GS10	GS11	Total
Day 1	0.647	5.713	2.267	8.627	1.033	10.301	3.910	15.243
	0.594	5.349	2.159	8.102	1.026	10.298	3.929	15.252
	0.603	5.382	2.176	8.161	1.056	10.501	3.984	15.540
	0.610	5.338	2.110	8.058	1.114	10.966	4.199	16.278
	0.630	5.589	2.265	8.483	1.060	10.317	3.983	15.360
	0.630	5.603	2.184	8.417	1.091	10.742	4.139	15.972
Day 2	0.572	5.562	2.358	8.491	0.989	10.001	3.673	14.663
	0.576	5.616	2.397	8.589	0.959	10.229	3.700	14.888
	0.540	5.215	2.224	7.979	1.018	10.408	3.775	15.201
	0.532	5.167	2.208	7.907	1.133	11.054	3.865	16.052
	0.549	5.280	2.304	8.134	1.052	10.476	3.769	15.297
	0.545	5.312	2.289	8.146	1.059	10.619	3.803	15.481
Day 3	0.574	6.229	2.242	9.046	1.103	11.351	4.014	16.469
	0.530	5.778	2.089	8.397	1.150	11.990	4.173	17.314
	0.570	6.051	2.155	8.777	1.172	11.757	4.048	16.977
	0.525	5.752	2.058	8.334	1.175	11.991	4.102	17.268
	0.546	5.916	2.117	8.578	1.097	11.406	3.945	16.448
	0.555	6.117	2.207	8.879	1.178	12.066	4.129	17.373
Average	0.57	5.61	2.21	8.39	1.08	10.92	3.95	15.95
St. Dev	0.04	0.32	0.09	0.32	0.06	0.68	0.16	0.87
%RSD	6.54	5.70	4.15	3.82	5.98	6.27	4.12	5.44
Confidence Interval	(0.56, 0.59)	(5.46, 5.76)	(2.17, 2.25)	(8.25, 8.54)	(1.05, 1.11)	(10.60, 11.23)	(3.88, 4.03)	(15.55, 16.35)

**Table 4 antioxidants-11-02441-t004:** Recovery percentages of accuracy test data in placebo samples (canola seeds, oil, and defatted meal).

	Recovery (%)									
		**Seed Matrix**			**Oil Matrix**			**Defatted Meal Matrix**		
Spike Level	Replicate	GS9	GS10	GS11	GS9	GS10	GS11	GS9	GS10	GS11
50%	1	101.86	95.41	95.10	101.45	101.59	103.05	101.44	103.94	97.52
	2	103.20	95.77	100.29	99.02	100.49	95.77	95.74	105.23	96.76
	3	102.84	95.25	95.49	100.24	100.62	94.43	102.82	103.02	95.69
	Average	102.64	95.48	96.96	100.23	100.90	97.75	100.00	104.07	96.66
150%	1	101.02	99.45	101.06	101.11	102.52	95.30	102.63	96.32	103.75
	2	100.58	99.63	99.89	100.57	105.81	98.02	102.56	99.68	105.10
	3	98.58	99.18	98.23	96.27	105.13	95.17	103.81	96.29	100.20
	Average	100.06	99.42	99.73	99.31	104.49	96.16	103.00	97.43	103.02

## Data Availability

Data is contained within the article.

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
