# Peer review of "Eco-Efficient Quantification of Glucosinolates in Camelina Seed, Oil, and Defatted Meal: Optimization, Development, and Validation of a UPLC-DAD Method"

_antioxidants, 2022, doi:10.3390/antiox11122441_

Round 1
Reviewer 1 Report
In the present work, the authors have developed a new UPLC method to extract and quantify total glucosinolates in camelina seed, oil, and defatted meal. The analytical steps of the method proposed is been designed properly. Every part of their analytical procedure seems solid and clear. Each variable has been thoroughly investigated and even the possible disadvantages of the use of the specific internal standard are discussed by them preempting possible comments. Overall, I believe that it is an excellent work that merits to be published.
Author Response
Response to Reviewer 1 Comments
Point 1: In the present work, the authors have developed a new UPLC method to extract and quantify total glucosinolates in camelina seed, oil, and defatted meal. The analytical steps of the method proposed is been designed properly. Every part of their analytical procedure seems solid and clear. Each variable has been thoroughly investigated and even the possible disadvantages of the use of the specific internal standard are discussed by them preempting possible comments. Overall, I believe that it is an excellent work that merits to be published.
Response 1: We appreciate the reviewer for their highly positive feedback.
Reviewer 2 Report
The manuscript entitled “Eco-efficient Quantification of Glucosinolates in Camelina Seed, Oil, and Defatted Meal: Optimization, Development, and Validation of a UPLC-DAD Method” developed and validated an eco-efficient UPLC-DAD method for determining the different Camelina glucosinolates, i.e. the glucoarabin (GS9), glucocamelinin (GS10), and homoglucocamelinin (GS11) in the camelina seed, oil, and defatted meal. The authors demonstrated that this method was suitable to be adopted as a new industrial standard for glucosinolate determination in camelina products.
Generally, the present paper was of importance for providing the new insights and strategies for the bioactive active components determination in the Camelina seed and the related products. However, this work is lack of innovation, and there are some points need to be addressed.
1. The abstract need to be improved. The authors used so many words to introduce the backgrounds of this work. The significance of the present work is also suggested to mention in the last sentence.
2. In the introduction part, the authors are suggested to share some words to introduce the research progress of the methods for quantification of glucosinolates in natural produces, and point to their shortcomings, thus concluded the innovative work that planned to be carried out here.
3. The resolution of the figures in the present work need to be improved, the fonts within the figures cannot be read clearly.
4. Which is more important among the factors that affected the efficiency of the glucosinolates quantification method, i.e. the HPLC machine, HPLC column, the gradient program of the mobile phase? I am also wonder whether the authors can get the same results if they do not use the Agilent HPLC machine, and the Waters HPLC column? This means that the authors are suggested to give the practical significance of their studies.
5. The conclusion parts should be improved. Some necessary information should be provided in this section.
Author Response
Response to Reviewer 2 Comments
The manuscript entitled “Eco-efficient Quantification of Glucosinolates in Camelina Seed, Oil, and Defatted Meal: Optimization, Development, and Validation of a UPLC-DAD Method” developed and validated an eco-efficient UPLC-DAD method for determining the different Camelina glucosinolates, i.e. the glucoarabin (GS9), glucocamelinin (GS10), and homoglucocamelinin (GS11) in the camelina seed, oil, and defatted meal. The authors demonstrated that this method was suitable to be adopted as a new industrial standard for glucosinolate determination in camelina products.
Generally, the present paper was of importance for providing the new insights and strategies for the bioactive active components determination in the Camelina seed and the related products. However, this work is lack of innovation, and there are some points need to be addressed.
Point 1: The abstract need to be improved. The authors used so many words to introduce the backgrounds of this work. The significance of the present work is also suggested to mention in the last sentence.
Response 1: We thank the reviewer for their constructive feedback. We have condensed the background (Line 12-13) in the abstract and further highlighted the significance of this work (Line 22-25).
Point 2: In the introduction part, the authors are suggested to share some words to introduce the research progress of the methods for quantification of glucosinolates in natural produces, and point to their shortcomings, thus concluded the innovative work that planned to be carried out here.
Response 2: We thank the reviewer for this suggestion. A mini-review of up-to-date and relevant glucosinolate methods has been moved from 3.1 to the Introduction section with minor modifications (Line 99-112). The novelties of our work have also been further highlighted (Line 113-123).
Point 3: The resolution of the figures in the present work needs to be improved, the fonts within the figures cannot be read clearly.
Response 3: We thank the reviewer for their feedback. We have revised all figures accordingly to improve the resolution. The original Figure 1 has been moved to the Supplementary document per the suggestion of another reviewer.
Point 4: Which is more important among the factors that affected the efficiency of the glucosinolates quantification method, i.e. the HPLC machine, HPLC column, the gradient program of the mobile phase? I am also wonder whether the authors can get the same results if they do not use the Agilent HPLC machine, and the Waters HPLC column? This means that the authors are suggested to give the practical significance of their studies.
Response 4: We thank the reviewer for this valuable question. We have added a paragraph (Line 306-311) to share our perspectives on using instruments from different brands. Regarding the overall efficiency gain that our UPLC method has shown, we believe it’s due to a combination of multiple factors that were discussed throughout 3.1 and 3.2.
Point 5: The conclusion parts should be improved. Some necessary information should be provided in this section.
Response 5: We thank the reviewer for this suggestion. We have completely revised the Conclusion section (Line 466-476).
Reviewer 3 Report
The aim of this study was develop and validate an eco-efficient UPLC-DAD method for extracting and quantifying glucosinolates in camelina seed, oil, and defatted meal.
The experimental work performed is suitable for the aim of the study, the paper is well organized and clear, and the state of the art of the topic is properly addressed. In addition, the English language and style are correct. However, some parameters need to be revised and/or improved before it being accepted for publication;
- The novelty of this work should be clearly stated in the Introduction.
- Keywords; "Antioxidant" must be deleted.
- Figure 1 must be included in Supplementary materials.
- The conclusion may be further improved, highlighting the main findings of the work (relevance, applicability).
Author Response
Response to Reviewer 3 Comments
The aim of this study was develop and validate an eco-efficient UPLC-DAD method for extracting and quantifying glucosinolates in camelina seed, oil, and defatted meal.
The experimental work performed is suitable for the aim of the study, the paper is well organized and clear, and the state of the art of the topic is properly addressed. In addition, the English language and style are correct. However, some parameters need to be revised and/or improved before it being accepted for publication;
Point 1: The novelty of this work should be clearly stated in the Introduction.
Response 1: We thank the reviewer for this valuable suggestion. The novelty of our work has been further highlighted in the Introduction (Line 113-123), Abstract (Line 22-25), and Conclusion sections (Line 466-467).
Point 2: Keywords; "Antioxidant" must be deleted.
Response 2: We thank the reviewer for their feedback. The Keywords have been revised accordingly (Line 26).
Point 3: Figure 1 must be included in Supplementary materials.
Response 3: We thank the reviewer for their feedback. The original Figure 1 has been moved to the Supplementary document as Figure 1S.
Point 4: The conclusion may be further improved, highlighting the main findings of the work (relevance, applicability).
Response 4: We thank the reviewer for this suggestion. We have completely rewritten the Conclusion section (Line 466-476).
Reviewer 4 Report
The paper is within the aims and the scope of the journal. Material definition must be improved. Methods are suitable, and used in a way that is possible to replicate experiments and analyses. The investigation is performed to good technical standards. It is no ethical problem involved. Discussion is sound and relevant. Conclusions should be improved.
Suggestions:
Line 30. Wrong botanical name of the plant. Instead of »Camelina Sativa« correct »Camelina sativa«.
Lines 115-117. Better derscription of samples are needed. The country of origion of camelina oil and canola samples. The location, year and conditions of growing plants?
Lines 449-452. Conclusions should be more informative.
Lines 504-505. Needed correction of the data of cited literature.
Lines 453-463. More information and description are needed to the supplementary data. For example, what is »Avg« in the table 7S.
Author Response
Response to Reviewer 4 Comments
The paper is within the aims and the scope of the journal. Material definition must be improved. Methods are suitable, and used in a way that is possible to replicate experiments and analyses. The investigation is performed to good technical standards. It is no ethical problem involved. Discussion is sound and relevant. Conclusions should be improved.
Point 1: Line 30. Wrong botanical name of the plant. Instead of »Camelina Sativa« correct »Camelina sativa«.
Response 1: We thank the reviewer for this suggestion. We have revised the wording accordingly (Line 29).
Point 2: Lines 115-117. Better description of samples is needed. The country of origin of camelina oil and canola samples. The location, year and conditions of growing plants?
Response 2: We thank the reviewer for this valuable feedback. Unfortunately, growing conditions of camelina seeds are currently undisclosable. The canola seeds were grown in a greenhouse in Saskatchewan, Canada in 2019.
Point 3: Lines 449-452. Conclusions should be more informative.
Response 3: We thank the reviewer for this suggestion. We have completely rewritten the Conclusion section (Line 466-476).
Point 4: Lines 504-505. Needed correction of the data of cited literature.
Response 4: We thank the reviewer for this feedback. The citation #13 has been corrected accordingly (Line 531-532)
Point 5: Lines 453-463. More information and description are needed to the supplementary data. For example, what is »Avg« in the table 7S.
Response 5: We thank the reviewer for this suggestion. The Abbreviations have been explained in the revised Supplementary document.
Round 2
Reviewer 2 Report
The authors have responded my comments and carefully made improvements about the previous version manuscript according to my suggestions. This improved version can be accepted after double checking language and grammars.